# High-dose naloxone: Effects by late administration on pain and hyperalgesia following a human heat injury model. A randomized, double-blind, placebo-controlled, crossover trial with an enriched enrollment design

**Anders Deichmann Springborg**[1]*, **Elisabeth Kjær Jensen**[1], **Mads Kreilgaard**[2], **Morten Aagaard Petersen**[3], **Theodoros Papathanasiou**[2], **Trine Meldgaard Lund**[2], **Bradley Kenneth Taylor**[4], **Mads Utke Werner**[1,5]

**1** Neuroscience Center, Copenhagen University Hospitals, Copenhagen, Denmark, **2** Department of Drug Design and Pharmacology, Faculty of Health and Medical Sciences, University of Copenhagen, Copenhagen, Denmark, **3** The Research Unit, Department of Palliative Medicine, Copenhagen University Hospitals, Copenhagen, Denmark, **4** Department of Anesthesiology, Pittsburgh Center for Pain Research, and the Pittsburgh Project to End Opioid Misuse, University of Pittsburgh, Pittsburgh, Pennsylvania, United States of America, **5** Department of Clinical Sciences, Lund University, Lund, Sweden

\* andersspringborg@gmail.com

## Abstract

Severe chronic postsurgical pain has a prevalence of 4–10% in the surgical population. The underlying nociceptive mechanisms have not been well characterized. Following the late resolution phase of an inflammatory injury, high-dose μ-opioid-receptor inverse agonists reinstate hypersensitivity to nociceptive stimuli. This unmasking of latent pain sensitization has been a consistent finding in rodents while only observed in a limited number of human volunteers. Latent sensitization could be a potential triggering venue in chronic postsurgical pain. The objective of the present trial was in detail to examine the association between injury-induced secondary hyperalgesia and naloxone-induced unmasking of latent sensitization. Healthy volunteers (n = 80) received a cutaneous heat injury (47°C, 420 s, 12.5 cm²). Baseline secondary hyperalgesia areas were assessed 1 h post-injury. Utilizing an enriched enrollment design, subjects with a magnitude of secondary hyperalgesia areas in the upper quartile ('high-sensitizers' [n = 20]) and the lower quartile ('low-sensitizers' [n = 20]) were selected for further study. In four consecutive experimental sessions (Sessions 1 to 4), the subjects at two sessions (Sessions 1 and 3) received a cutaneous heat injury followed 168 h later (Sessions 2 and 4) by a three-step target-controlled intravenous infusion of naloxone (3.25 mg/kg), or normal saline. Assessments of secondary hyperalgesia areas were made immediately before and stepwise during the infusions. Simple univariate statistics revealed no significant differences in secondary hyperalgesia areas between naloxone and placebo treatments (P = 0.215), or between 'high-sensitizers' and 'low-sensitizers' (P = 0.757). In a mixed-effects model, secondary hyperalgesia areas were significantly larger following

**Data Availability Statement:** All relevant data are within the manuscript and its Supporting Information files.

**Funding:** The trial received financial support by Aase og Ejnar Danielsens Fond (https://danielsensfond.dk/) grant number 10-001534 to MUW, Brødrene Hartmanns Fond (https://www.hartmannfonden.dk/) grant number A28468 to MUW, Augustinus Fonden (https://augustinusfonden.dk/) grant number 15-1724 to MUW, and the United States National Institutes of Health (National Institute on Drug Abuse [NIDA]; https://www.drugabuse.gov/) grant number R01DA037621 to BKT and MUW. The funders had no role in study design, data collection and analysis, decision to publish, or preparation of the manuscript.

**Competing interests:** The authors have declared that no competing interests exist.

naloxone as compared to placebo for 'high-sensitizers' (P < 0.001), but not 'low-sensitizers' (P = 0.651). Although we could not unequivocally demonstrate naloxone-induced reinstatement of heat injury-induced hyperalgesia, further studies in clinical postsurgical pain models are warranted.

## Introduction

The endogenous opioid analgesia system can be impaired or altered in chronic pain conditions [1–4], playing a putative pathophysiological role in the transition from acute to chronic pain [5–7]. Naloxone and naltrexone are μ-opioid-receptor (MOR) inverse agonists [6] used in experimental research to determine the activity of the endogenous opioid analgesia system [8, 9]. Naloxone produces either hypoalgesic or hyperalgesic responses to nociceptive stimulation, depending on the dose administered [10]. Studies in rodents indicate that endogenous MOR constitutive activity masks a process called latent sensitization [6], defined as an increased responsiveness of nociceptive neurons to afferent input induced by either injury, chronic opioid administration, or physiological stress. Latent sensitization can persist in the absence of behavioral signs of hypersensitivity and outlasts the duration of tissue healing. It can be revealed upon administration of an opioid receptor inverse agonist, leading to reinstatement of hyperalgesia in rodents [5, 6, 11, 12].

In an initial randomized, controlled, crossover trial design, we reported that an intravenous dose of naloxone (21 microg/kg), delivered 72 h after a cutaneous heat injury (CHI), failed to reinstate hyperalgesia in healthy human subjects [13]. In a follow-up trial using a higher dose of naloxone (2 mg/kg), four out of twelve subjects demonstrated reinstatement of hyperalgesia [14]. Reinstatement of latent sensitization, if consistently present in humans may constitute one of the basic trigger mechanisms in development of chronic pain states, e.g. chronic postsurgical pain.

Furthermore, we noticed that the subjects developing reinstatement of hyperalgesia were subjects with larger initial areas of secondary hyperalgesia (SHA), leading us to hypothesize that latent sensitization occurs more often in 'high-sensitizers' as compared to 'low-sensitizers'. If the hypothesis is validated, the enriched design could be used in future trials to determine indicators of vulnerability to chronic postsurgical pain [14].

The objectives of the current trial were, first, to replicate [15] our previous latent sensitization trial applying a larger sample size. Second, using an enriched enrollment design, to examine whether 'high-sensitizers' express larger hyperalgesia areas after a naloxone challenge than 'low-sensitizers'.

## Materials and methods

### Trial management

The trial was approved by the Committee of Health Research Ethics of the Capital Region (H-15018869), the Danish Medicines Agency (2015–005426–19), and the Data Inspection Authority of the Capital Region (RH-2015-284, I-suite no. 04296). Trial registrations were additionally in EUDRACT (2015-005426-19, registered on January 22, 2016) and ClinicalTrials.gov (NCT02684669, registered on February 10, 2016), with the principal investigator MUW. The trial protocol with detailed methodological information has been published [16] and the original approved protocol and the CONSORT checklist are available as supporting information (S1 Protocol; S1 Checklist).

## Participants

Participants were recruited from a registry of previously completed experimental pain studies at the Neuroscience Center, Copenhagen University Hospitals and by an advertisement at the Danish website forsoegsperson.dk (Inclusion and exclusion criteria Table 1). Additionally, the advertisement was posted on facebook.com and hung up on bulletin boards at the University of Copenhagen. Following written and verbal information signed informed consents were obtained from all subjects prior to any assessments. The enrollment process was performed by the corresponding author (ADS). The CONSORT flow diagram shows the included subjects in each session of the trial (Fig 1). The complete date range for participant recruitment and follow-up was February 22, 2016 to October 1, 2016.

Females were not allowed to participate in the study, since it cannot be excluded that exposure to the supra-pharmacological dose (high-dose) of naloxone may cause teratogenic effects: pregnancy tests are not reliable indicators of a gestational age of less than five weeks. The pregnancy prevention measures, e.g. mechanical or hormonal, have an anti-conception success rate below 100%.

## Laboratory environment

The experimental procedures took place at the Multidisciplinary Pain Center, Neuroscience Center, Copenhagen University Hospital, and were performed in a quiet, daylighted room (22–25˚C; relative humidity 20%–45%). Participants adopted a relaxed, recumbent position during sensory assessments. Sensory assessments were performed between 8:00 AM and 4:00 PM.

## Trial design

A randomized, double-blinded, placebo-controlled, crossover trial with an enriched design.

**Table 1. Inclusion and exclusion criteria.**

| Inclusion criteria | Exclusion criteria |
|---|---|
| Healthy male | Participant does not speak or understands Danish |
| Age above 18 years and below 35 years | Participant cannot cooperate with the investigation |
| Signed informed consent | Allergic reaction against morphine or other opioids (incl. naloxone) |
| Urine sample without traces of opioids | Alcohol or drug abuse |
| ASA I | Use of psychotropic drugs (exception of SSRI) |
| Body mass index: $18 < BMI < 30$ kg/m$^2$ | Neurologic or psychiatric disease |
| | Signs of neuropathy in the examination region |
| | Previous severe trauma to the lower legs with sequelae |
| | Scarring or tattoos in the test region |
| | Chronic pain condition |
| | Regular use of analgesic drugs |
| | Use of prescription drugs one week before the trial |
| | Use of over-the-counter drugs 48 hours before the trial |
| | Does not develop measurable secondary hyperalgesia areas after the mild heat injury |

**ASA**: American Society of Anesthesiology's physical status classification system; **BMI**: Body mass index; **SSRI**: Selective serotonin reuptake inhibitors.

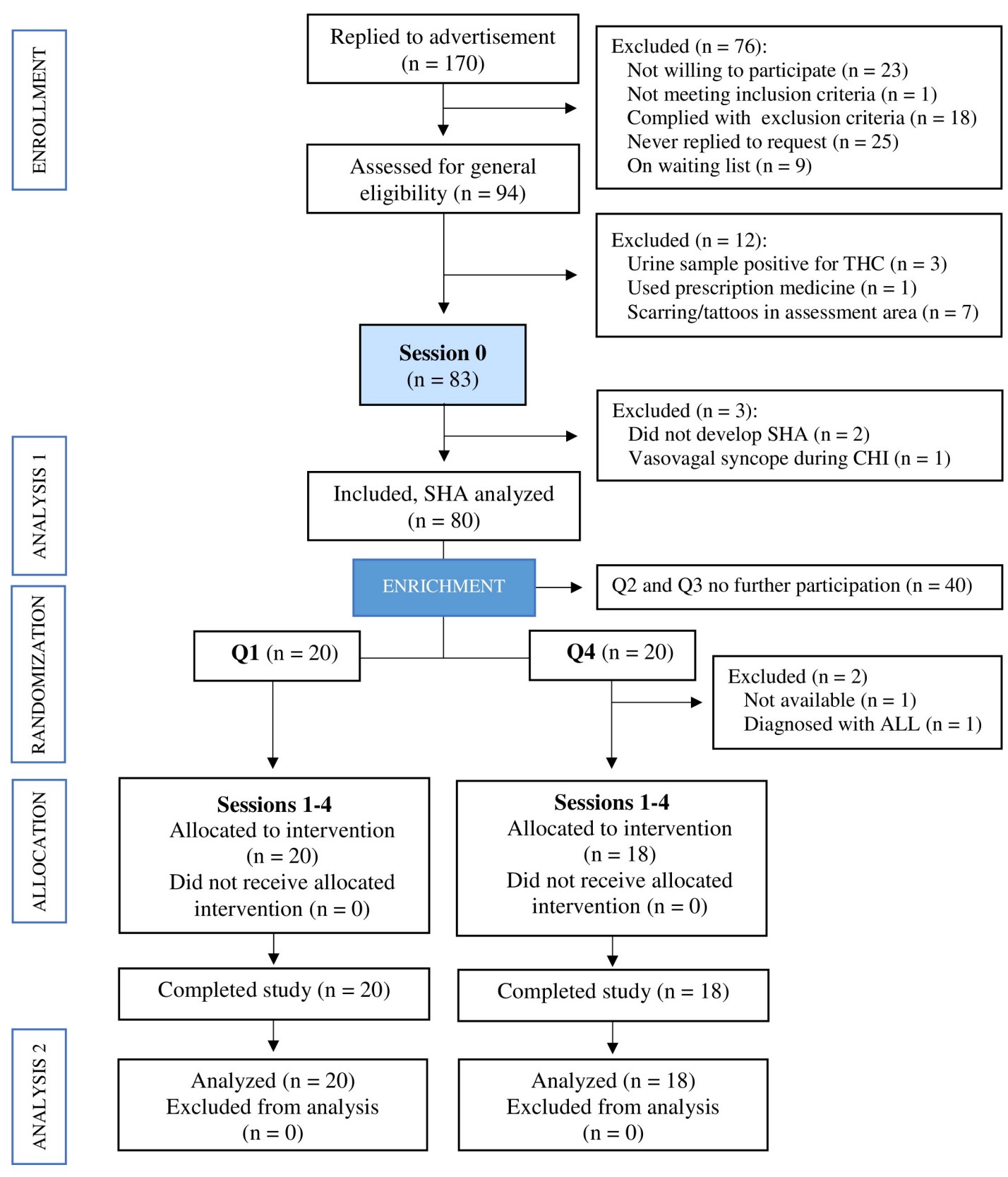

**Fig 1. CONSORT flow diagram.** A total of 170 individuals replied to the advertisement. Ninety-four subjects were assessed for general eligibility on an enrichment selection session (Session **0**). Twelve subjects were excluded before the assessments, three tested positive for tetrahydrocannabinol (THC) in their urine, one used prescription drugs and seven had scars or tattoos in the assessment areas. Eighty-three subjects received a cutaneous heat injury (CHI) on Session **0**. Subsequently, three subjects were excluded, two subjects did not develop secondary hyperalgesia areas (SHAs), and one subject had a vasovagal syncope following the CHI. Eighty subjects' SHAs were analyzed and ranked according to magnitude, assessed by planimetric measurements. Through an enrichment process, 40 subjects with SHAs in the two middle quartiles (Q2/Q3) were discontinued, while the 20 subjects belonging to the lower quartile (Q1) and the 20 subjects belonging to the upper quartile (Q4) were randomized and continued to the experimental sessions. Two subjects from Q4 were, however, excluded before the allocation procedure in Session **1** for reasons unrelated to the trial, one subject was unavailable, and another was diagnosed with acute lymphocytic leukemia (ALL). All other subjects completed per-protocol the trial sessions, and the final analysis, thus, included 38 subjects.

**Enrichment.** The trial included an enrichment selection session (Session **0**) and four experimental sessions (Sessions **1** to **4**; Fig 2). The enrichment enrollment selection [17, 18] was based on the magnitude of SHA following a CHI, thus separating 'high-sensitizers' (upper quartile [Q4]) and 'low-sensitizers' (lower quartile [Q1]), from 'intermediate-sensitizers' (Q2/Q3), who were excluded from further trial participation. Subjects in Q4 (n = 20) and Q1 (n = 20) continued to experimental Sessions **1** to **4**. However, two subjects from Q4 were excluded before the allocation procedure in Session **1** for reasons unrelated to the study (Fig 1).

**Randomization procedure.** After Session **0** a computer-generated random permutation of numbers 1 to 40 was applied to subjects in Q1 (n = 20) and Q4 (n = 20; randomization. com). Each subject from Q1 and Q4 then received a sequential rank order according to the magnitude of the SHA (1 to 20; 61 to 80). These sequential rank numbers were then consecutively paired with the list of random permutation of numbers, rendering randomization of the order of 'high- and low-sensitizers'. The subjects were then invited to participate in the experimental sessions in this randomized order. This randomization procedure was performed by the principal investigator (MUW; not participating in the assessments), and source data were locked away safely. The examiners were blinded to the subject´s 'sensitizer'-affiliation.

Computer-Generated sequence randomization, using blocks of four subjects (randomization.com), was performed by the hospital pharmacy (Skanderborg Pharmacy, Skanderborg, DENMARK), responsible for manufacturing, labeling and packaging of the drugs. Two sets of non-transparent, sealed envelopes, containing information on treatment

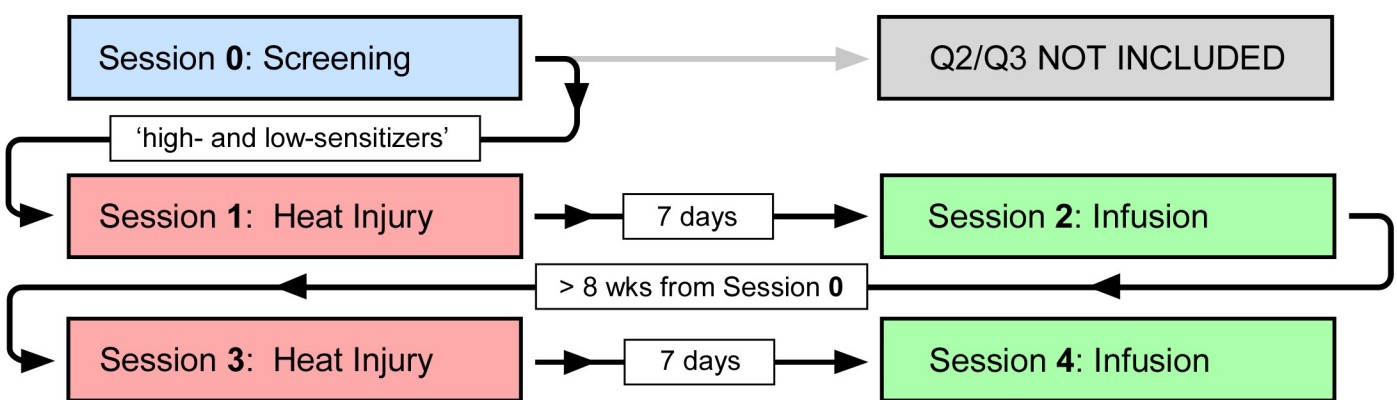

**Fig 2. General trial layout.** Cutaneous heat injuries were induced on an enrichment session (Session **0**), a selection process uncovering 'high-sensitizers' (large secondary hyperalgesia areas: upper quartile [Q4]) and 'low-sensitizers' (small secondary hyperalgesia areas: lower quartile [Q1]). Sessions **1** and **3** included repeat cutaneous heat injuries in 'high- and low-sensitizers'. Target-controlled infusion sessions were Sessions **2** and **4**, with randomized allocation between placebo and naloxone. The time interval between Sessions **1** and **2**, and, Sessions **3** and **4**, was 7 days. The time interval between Sessions **0** and **3** was > 8 weeks.

allocation order for each participant, were prepared and stored securely while an additional envelope was kept in the examination room, to be opened in case of a medical emergency. Both participants and investigators were blinded to the treatment sequence throughout the trial.

## Cutaneous heat injury

The CHIs were induced on the thigh (Session **0**) and the medial aspect of the calf (Session **1**: right calf; Session **3**: left calf) with a computerized contact thermode system (MSA Thermal Stimulator, Somedic AB, Hörby, SWEDEN; heating area: 2.5 x 5.0 cm2; baseline: 32˚C; ramp rate: ± 1˚C/s; plateau: 47˚C; duration: 420 seconds) [13, 19, 20]. The homotopic testing areas were meticulously delineated at each session.

## Drug administration

In Sessions **2** and **4**, target-controlled infusions (TCI) of naloxone or placebo were administered 168 h after induction of the CHI (Fig 3). The TCI-algorithm, based on previously reported population pharmacokinetic data [21], was calculated by the software NONMEM (7.3 ICON Development Solutions, Manchester, U.K. [property of UCSF, CA]), using computer simulations based on a population-kinetic model with 2,000 simulated administrations distributed on ten subjects. The estimated mean (10% and 90% percentiles) plasma concentrations of naloxone at each of the three TCI-steps are illustrated in Fig 4 [22] with each TCI-step containing a 1 min bolus and a 24 min continuous infusion. A total dose of naloxone 3.25 mg/kg (4 mg/mL) vis-á-vis normal saline 0.81 mL/kg was administered (Table 2). During the last 10 min of each 25 min step, plasma concentrations were considered to be stable, and therefore sensory assessments were performed (Fig 4).

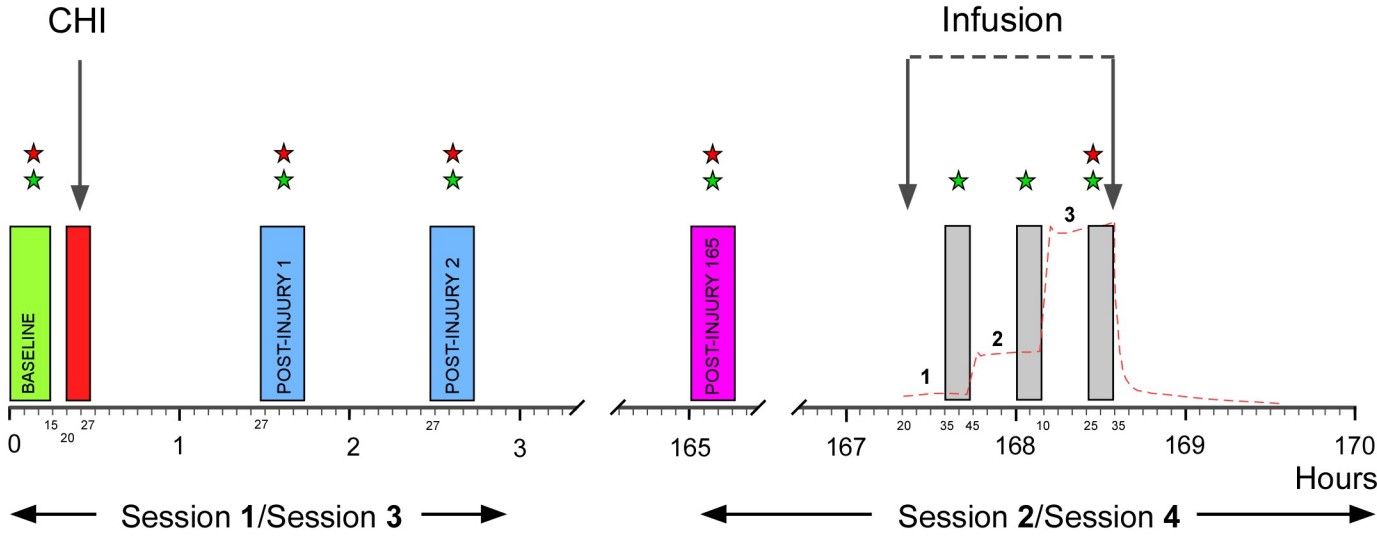

**Fig 3. Trial timeline.** Session **1** and **3** (cf. Fig 1) included baseline assessments (green rectangle, 0 min), induction of a cutaneous heat injury (CHI, red rectangle, 20 min) and post-injury assessments (blue rectangles: 1 h 27 min and 2 h 27 min). Session **2** and **4** (cf. Fig 1) included a pre-drug assessment (magenta rectangle; post-injury 165 h), drug-infusions (naloxone or placebo; 167 h 35 min, 168 h 0 min, and 168 h 25 min), and assessments during target-controlled infusion (TCI; grey rectangles; 167 h 35 min, 168 h 0 min, and 168 h 25 min). The estimated TCI plasma concentrations are superimposed in dashed red line. Numbers (1 to 3) during the infusion period, indicate the three TCI-steps. Assessments included secondary hyperalgesia areas and online reaction time indicated by green stars and pin-prick pain thresholds indicated by red stars.

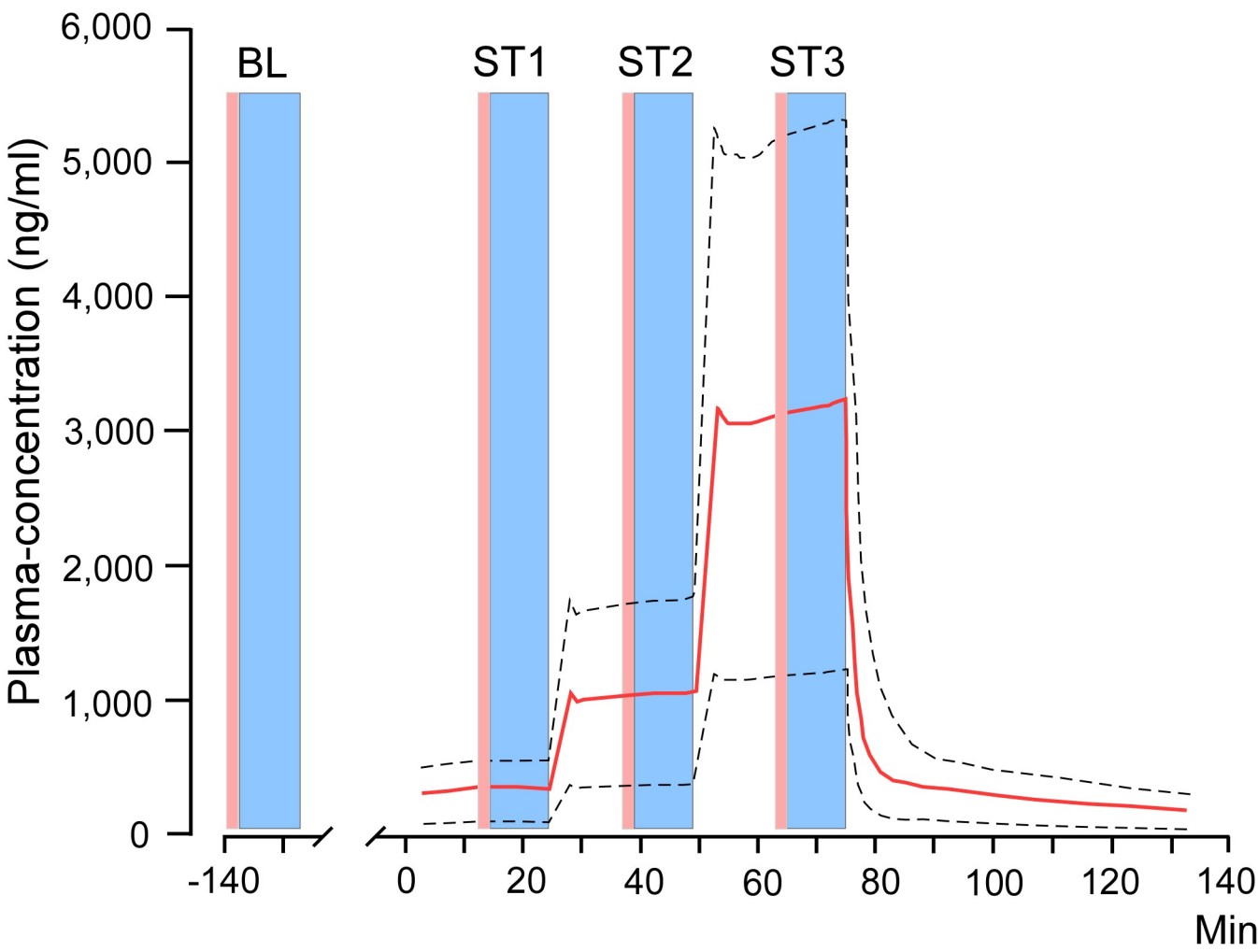

**Fig 4. Target-controlled infusion.** Test-algorithm Session **2** and **4** for subjects with superimposed naloxone plasma-concentration curves. Median plasma-concentration (red) with 10% and 90% percentiles ranges (dashed black lines) during a three-step target-controlled infusion (TCI). Naloxone was administered at step 1 (ST1: 15 min to 25 min) with 0.25 mg/kg, step 2 (ST2: 39 min to 49 min) 0.75 mg/kg, and step 3 (ST3: 65 min to 75 min) 2.25 mg/kg. Magenta columns represent ratings with Clinical Opiate Withdrawal Scale (COWS) and blue columns indicate sensory testing (BL and ST1 to ST3: secondary hyperalgesia areas; BL and ST3: pin-prick pain thresholds). **BL**: baseline assessments [22].

**Monitoring.** Vital sign monitoring in TCI-sessions (Sessions **2** and **4**) was with continuous three-lead electrocardiogram, heart rate, pulse oximetry, noninvasive arterial blood pressure, and respiratory rate.

## Primary outcome

**Secondary hyperalgesia areas.** Areas of secondary hyperalgesia were assessed by a weighted-pin instrument (128 mN; MRC Systems, Heidelberg, GERMANY) [20] delivering punctate stimuli (stimulus area: 0.049 mm$^2$) in Sessions **0**, **1**, and **3**: before, and 1 and 2 h after the CHI (Fig 2); and Sessions **2** and **4**: 165 to 169 h after the CHI (baseline: 165 h 0 min; during TCI: 167 h 35 min, 168 h 0 min, and 169 h 25 min). The borders of the SHAs were determined by stimulating along eight symmetric lines starting in normal skin and converging towards the center of the

**Table 2. Target-controlled infusion (TCI) of naloxone.**

| Step | TCI | Time (min) | Dose (mg/kg) |
|---|---|---|---|
| **Step 1** | Bolus 1 | 0–1 | 0.02 |
| | Infusion 1 | 1–25 | 0.23 |
| **Step 2** | Bolus 2 | 25–26 | 0.06 |
| | Infusion 2 | 26–50 | 0.69 |
| **Step 3** | Bolus 3 | 50–51 | 0.18 |
| | Infusion 3 | 51–75 | 2.07 |
| | TOTAL | 75 | 3.25 |

Target-controlled infusion (TCI) of naloxone (4 mg/ml): timeline and dose/kg for the three steps: **Step 1** (Bolus 1 + Infusion 1), **Step 2** (Bolus 2 + Infusion 2), and **Step 3** (Bolus 3 + Infusion 3).

CHI. The participant reported when the punctate sensation changed from an innocuous pin-prick to a stinging, smarting, or unpleasant sensation. The corners of the octagon were marked on the skin and transferred to a clear acetate sheet. The SHA was determined by planimetrics using a vector-based drawing program (Canvas 12.0, ACD Systems International, BC, CANADA).

## Other outcomes

**Mechanical pain thresholds.** Pin-prick pain thresholds (PPTs) were assessed in the primary and secondary injury area by punctate stimulators (8, 16, 32, 64, 128, 256, and 512 mN) using a modified Dixon procedure (Fig 3) [23, 24]. Pin-prick pain thresholds were assessed in Sessions **0**, **1**, and **3** (baseline, 1, and 2 h post-injury) and Sessions **2** and **4** (before the infusion 165 h 0 min and during Step 3 of the TCI 168 h 25 min). The punctate stimulators were applied perpendicularly to the skin, each five times. The lowest nominal value of the punctate stimulator eliciting $\geq 3$ stinging, smarting, or unpleasant sensations indicated the PPT. The median value of the four PPTs was used for further analysis. The PPT data was converted to corresponding ordinal values (1 to 8).

**Pain during the cutaneous heat injury.** During the CHI, pain assessments (Numerical Rating Scale [NRS]; 0 to 10) were made at 0, 30, 60, 120, 180, 240, 300, 360, and 420 s. The CHI-induced pain intensity was calculated using area under the curve per second (AUC/s).

**Clinical opiate withdrawal scale.** Clinical signs of endogenous opiate withdrawal during the naloxone infusion were assessed by the examiner using the Clinical Opiate Withdrawal Scale (COWS) [25] on Session **0**, and during the TCI-infusion on Sessions **2** and **4** (Fig 4).

**Psychometric evaluations.** Since anxiety and depression are well-known psychological parameters intimately associated with pain, participants completed the Hospital Anxiety and Depression Scale (HADS-A/D) [26, 27], and the Pain Catastrophizing Scale (PCS) [28, 29] at Session **0** before the sensory assessments.

**Online reaction time test.** Reaction time was tested since it is a measure of vigilance, important during sensory testing. Further, the possible sedating effects of high-dose naloxone was evaluated. Online reaction time was assessed using a computer application showing a red-green traffic light (https://faculty.washington.edu/chudler/java/redgreen.html; Fig 3) [30]. Participants were asked to press a button immediately when the light switched from red to green. The median of three measurements indicated the reaction time.

## Statistics

**Sample size calculation.** Data from a previous high-dose naloxone trial (n = 12) [14] in 'high-sensitizers' (n = 3), 168 h post-injury, indicated mean (SD) values of SHAs during

naloxone infusion of 111.0 cm2 (26.3 cm2) and during placebo infusion 2.1 cm2 (2.5 cm2). Correspondingly for 'low-sensitizers' (n = 3), SHAs were 0.9 cm2 (0.6 cm2) and 0.3 cm2 (0.1 cm2), respectively. With a significance level of 0.01 (α) and a power of 0.90 (β = 0.10), the estimated number of individuals needed to reject the null hypothesis in 'high-sensitizers' were 5 (effect size 4.3) and in 'low-sensitizers' 18 (effect size 1.1; G*Power3.9.1.2, Kiel University, GERMANY). However, because the sample size estimate was based on data with excessive variability, it was decided to include 20 'high-sensitizers' and 20 'low-sensitizers'.

**Statistical analysis.** Data distributions were inferred from residual plots and the Kolmogorov–Smirnov test (SPSS IBM Software 22.0, Chicago, IL; MedCalc Software: version 16.4.3, Mariakerke, BELGIUM). The basic SHA-arithmetic was (NX = naloxone; PLA = placebo):

$$\Delta SHA_{Q4} = SHA_{NX\ Q4} - SHA_{PLA\ Q4}$$

$$\Delta SHA_{Q1} = SHA_{NX\ Q1} - SHA_{PLA\ Q1}$$

$$\Delta SHA = \Delta SHA_{Q4} - \Delta SHA_{Q1}$$

The main analysis used maximal SHA values ($SHA_{MAX}$) during the TCI (independent of TCI-step).

The primary outcome SHA was analyzed according to the protocol (S1 Protocol) [16], by simple univariate statistics (Wilcoxon signed-rank test, Mann-Whitney, paired or unpaired t test) and by an advanced mixed-effects model. The mixed-effects model was with a random effect for subject, and fixed effects for the variables 'sensitizers' ('high-sensitizers'/'low-sensitizers'), intervention (naloxone/placebo), TCI-step (step 1, 2, 3), time (Session **2**, **4** [added post hoc, see Results—Protocol violations for explanation]), HADS-scores, and PCS-scores for the primary outcome measure SHA. The starting model included all interactions. Main effects and interaction effects were examined. Non-significant factors ($P > 0.05$), beginning with interactions, were excluded until all included factors attained significance. Carry-over effects were assessed using SHA data from the CHI-sessions (Sessions **1** and **3**).

Other outcomes were analyzed by univariate statistics (PPTs, pain during the CHI), ICCs (pain during the CHI), and two-way repeated-measures ANOVA (online reaction time). A P-value $< 0.01$ was considered statistically significant. Data are given as mean (95% CI) or median (95% CI), as appropriate.

**Validity of the enrichment design.** To test the validity of the enrichment design (the agreement across Sessions **0**, **1**, and **3**), ICCs (two-way random model with absolute agreement), and one-way repeated-measures ANOVA or the Friedman test pertaining SHA-data (Session **0**, **1**, and **3**) were calculated. For interpretation purposes, ICCs were categorized as slight/poor ($< 0.2$), fair (0.2 to 0.4), moderate (0.4 to 0.6), substantial (0.6 to 0.8), and almost perfect ($> 0.8$) [31].

## Results

### Volunteers and trial chronology

A total of 94 subjects were assessed for eligibility (CONSORT flow diagram Fig 1), and 83 subjects were enrolled in Session **0** assessments. Three of these subjects were excluded, and the data from the remaining 80 subjects were analyzed per-protocol [16] (data available as supporting information: S1 Table). Forty subjects, 'high-sensitizers' (Q4; n = 20) and 'low-sensitizers' (Q1; n = 20), continued on to Sessions **1** to **4**. Two Q4-subjects were excluded for reasons not related to the trial (Fig 1; specified in S1 Adverse events). Per-protocol data from the

**Table 3. Demographics and anthropometrics.**

|  | Included (Q4/Q1) (n = 40) | Final analyzed (Q4/Q1) (n = 38) | Excluded (Q2/Q3) (n = 40) |
|---|---|---|---|
| **Age (yrs)** | 23.6 (22.6–24.7) | 23.6 (22.6–24.6) | 23 (22.0–24.0)[§] |
| **Height (cm)** | 182.6 (180.3–184.8) | 182.8 (180.6–185.1) | 182.5 (180.3–184.6) |
| **Weight (kg)** | 79.1 (75.4–82.9) | 80.2 (76.6–83.8) | 79.3 (75.7–82.9) |
| **BMI (kg/m$^2$)** | 23.7 (22.8–24.6) | 24.0 (23.1–24.9) | 23.8 (22.9–24.6) |
| **BSA (m$^2$)** | 2.00 (1.94–2.05) | 2.01 (1.96–2.07) | 2.00 (1.95–2.05) |

Demographics and anthropometrics for the included group (First quartile [Q1] and fourth quartile [Q4] [see text for explanation]), the final analysis group (Q1/Q4) and the excluded group (Q2/Q3). Values are presented as mean (95% CI) unless otherwise indicated. [§] median (95% CI). **BMI**: body mass index; **BSA**: body surface area.

remaining 38 subjects were analyzed. Demographic and anthropometric data are presented in Table 3. The first and last trial visits were February 29, 2016 and September 30, 2016, respectively. The time intervals, median (95% CI), between Sessions **0** and **1**, and, Sessions **1** and **3**, were 60 (45–76) days and 47 (41–54) days, respectively.

## Methodology

**Changes induced by the cutaneous heat injury.** The CHI induced significant SHAs in both Sessions **1** and **3** (Friedman tests: P < 0.001) and all subjects had measurable SHAs in both sessions. Furthermore, the CHI significantly reduced PPTs in both the primary and secondary areas of hyperalgesia in Sessions **1** and **3** (Friedman tests: P < 0.001).

**Validity of the enrichment procedure.** The ICCs (95% CI) for the SHA data from the CHI-sessions (Sessions **0**, **1**, and **3**) for Q1 (n = 20), Q4 (n = 18), and Q1 and Q4 combined (n = 38) were 0.59 (0.13–0.82; P = 0.010), 0.58 (0.09–0.83; P = 0.015), and 0.72 (0.52–0.85; P < 0.001), respectively, indicating moderate to substantial agreement across sessions (Fig 5). A Friedman test showed differences in SHAs between Sessions **0**, **1**, and **3** for Q1 (Chi-square [df]: 13.3 [2], P = 0.001) but not for Q4 (8.1 [2], P = 0.017). Further analysis revealed that SHAs at Session **1** (median [95% CI]: 27.2 [23.9–33.6] cm$^2$) was significantly larger than at Session **0** for Q1 (median [95% CI]: 21.2 [19.2–23.9] cm$^2$; Wilcoxon signed-rank test: -3.44, P = 0.001).

**Carry-over effects.** A trend towards larger SHAs in Session **1** (median [95% CI]: 34.2 [27.2–44.6] cm$^2$) compared to Session **3** (median [95% CI]: 28.7 [24.0–42.4] cm$^2$; Wilcoxon signed-rank test: P = 0.016), was observed.

## Primary outcome

**Pairwise comparison.** When analyzing SHA$_{MAX}$ without partitioning into 'high-sensitizers' and 'low-sensitizers' there was no significant difference between naloxone (median [95% CI]: 0 [0–0.3] cm$^2$) and placebo (median [95% CI]: 0 [0–0] cm$^2$; Wilcoxon signed-rank test: P = 0.215; Fig 6). In the 'high-sensitizer' group, no significant difference was demonstrated between naloxone (median [95% CI]: 0 [0–75.5] cm$^2$) and placebo (median [95% CI]: 0 [0–11.7] cm$^2$; Wilcoxon signed-rank test: P = 0.374). Similarly, in 'low-sensitizers', no significant difference was demonstrated between naloxone (median [95% CI]: 0 [0–0] cm$^2$) and placebo (median [95% CI]: 0 [0–0] cm$^2$; Wilcoxon signed-rank test: P = 0.398). No statistical difference for ΔSHA$_{MAX}$ was found between 'high-sensitizers' (median [95% CI]: 0 [0–14.4] cm$^2$) and 'low-sensitizers' (median [95% CI]: 0 [0–0] cm$^2$; Mann-Whitney U test: P = 0.757).

**Mixed-effects model.** No interactions attained significance. Main effects of intervention (P = 0.015), 'sensitizers', TCI-step, HADS-scores, and PCS-scores were not significantly

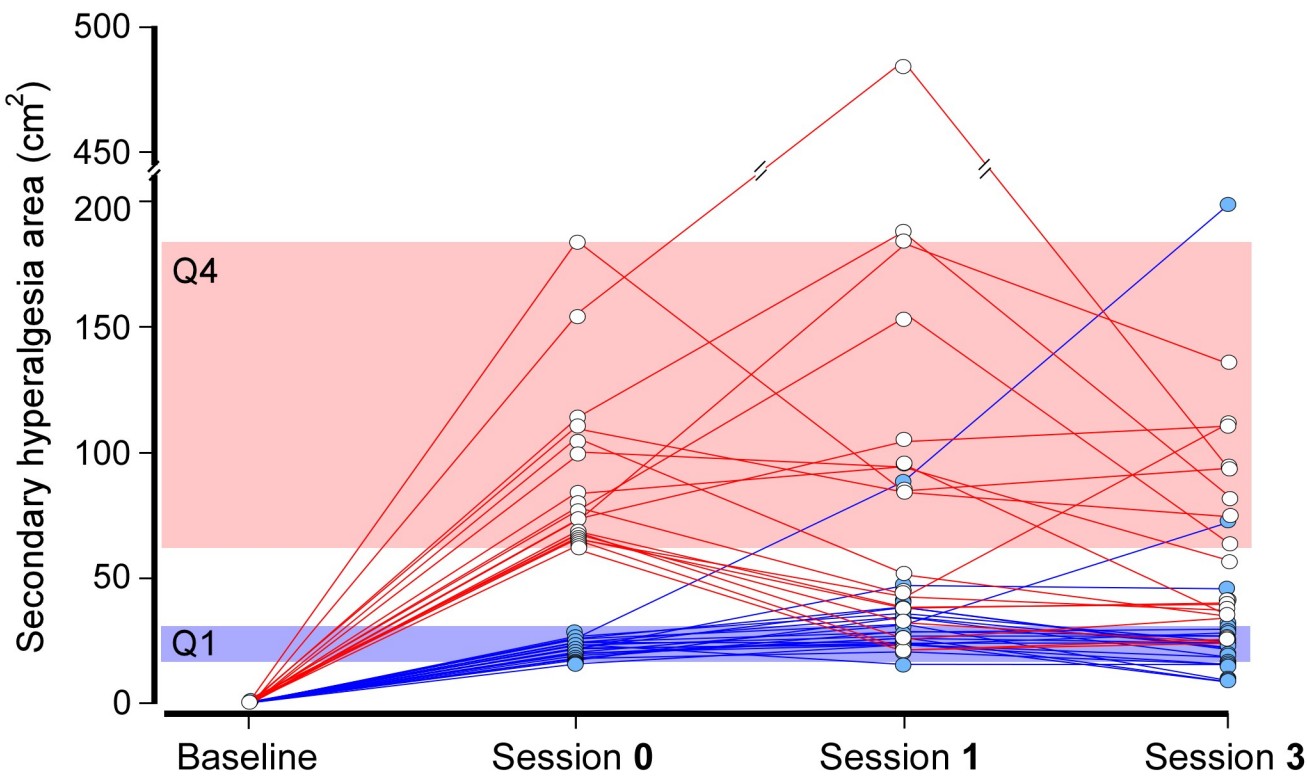

**Fig 5. Sensitizer categorization following cutaneous heat injuries.** Secondary hyperalgesia areas as measured at baseline on Sessions **0**, **1**, and **3**. Red lines illustrate 'high-sensitizers' and blue lines illustrate 'low-sensitizers'. The blue rectangle shows the Q1 interval and the red rectangle shows the Q4 interval as defined on Session **0**.

different (Table 4). In the post hoc analysis model (cf. Materials and methods—Statistical analysis; Table 5), where the variable time was included, a significant interaction between intervention*'sensitizers' (P = 0.002), as well as between 'sensitizers'*time (P < 0.001) was observed. Regarding the estimates of fixed effects, SHA following naloxone compared to placebo was 35.4 cm$^2$ (P < 0.001; Table 6) for 'high-sensitizers'. However, this difference was not seen in 'low-sensitizers' (P = 0.651). Main effects for TCI-step, HADS-scores, and PCS-scores were not significantly different (Table 5).

## Other outcomes

**Mechanical pain thresholds.** There was no significant difference between PPTs in the area of primary hyperalgesia following naloxone (median [95% CI]: 0 [0–0]) compared to placebo (median [95% CI]: 0 [0–0]; Wilcoxon signed-rank test: P = 0.663) when correcting for baseline by using the difference between pre-infusion and post-infusion PPT values. Furthermore, no significant difference was seen for PPTs in the area of secondary hyperalgesia following naloxone (median [95% CI]: 0 [0–0]) compared to placebo (median [95% CI]: 0 [0–0]; Wilcoxon signed-rank test: P = 0.854).

**Pain during induction of the cutaneous heat injury.** There was no significant difference in pain intensity during Session **1** (mean [95% CI]: 4.2 [3.6–4.8] NRS) compared to Session **3** (mean [95% CI]: 4.2 [3.6–4.8] NRS; Paired t test: P = 0.808). Almost perfect inter-session

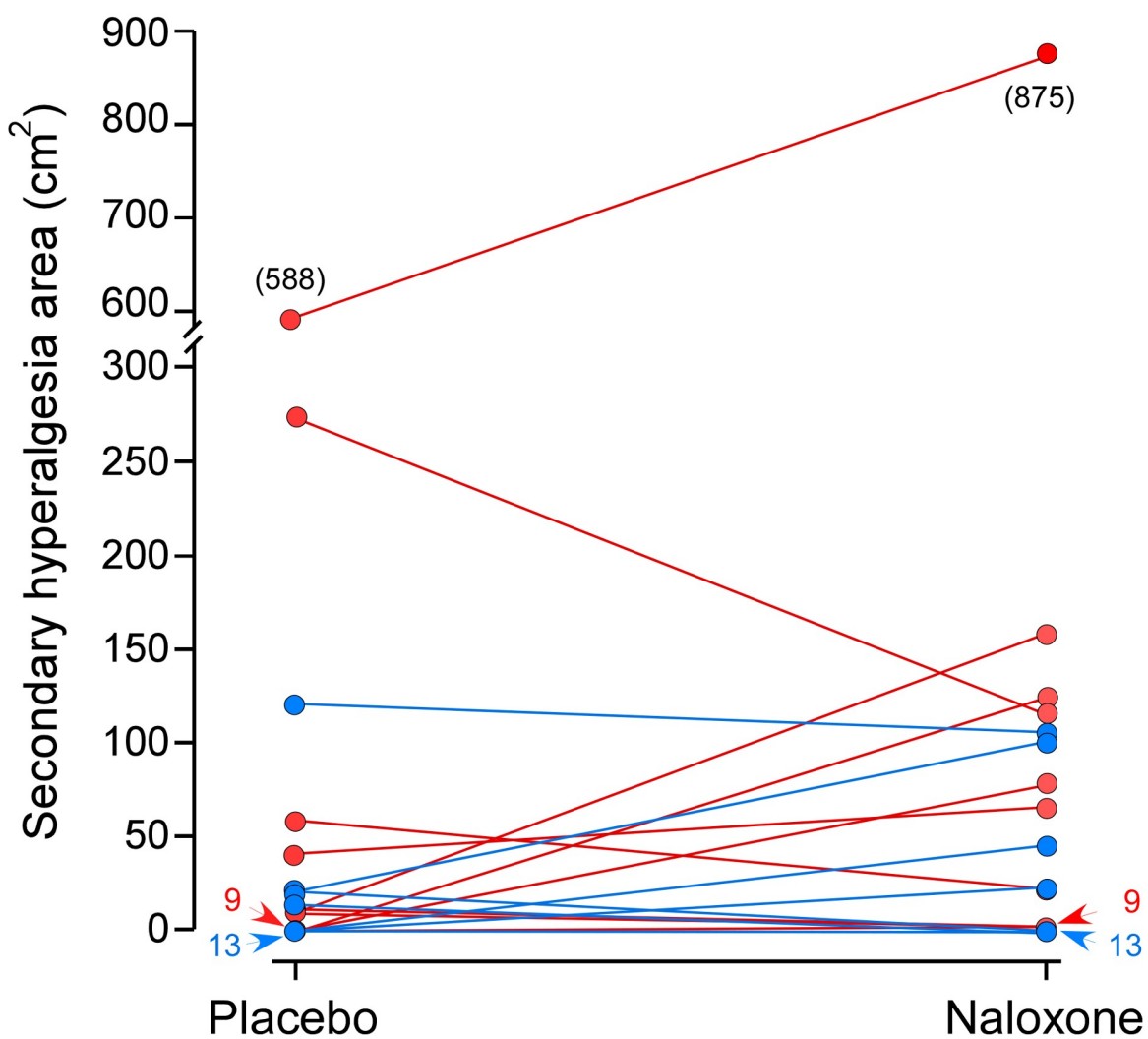

**Fig 6. Magnitude of secondary hyperalgesia areas following naloxone or placebo.** Comparison of individual maximum secondary hyperalgesia area ($SHA_{MAX}$) with subtraction of baseline areas after administration of naloxone or placebo. When all subjects were analyzed without partitioning into 'high- and low-sensitizers', there were no significant difference between SHAs following naloxone infusion (median [95% CI]: 0 [0–0.3] $cm^2$), compared to the placebo infusion (median [95% CI]: 0 [0–0] $cm^2$; Wilcoxon signed-rank test: P = 0.215). Furthermore, no significant difference for $SHA_{MAX}$ was found between 'high-sensitizers' (median [95% CI]: 0 [0–14.4] $cm^2$) and 'low-sensitizers' (median [95% CI]: 0 [0–0] $cm^2$; Mann-Whitney U test: P = 0.757). Red lines represent 'high-sensitizers' and blue lines represent 'low-sensitizers'.

consistencies in pain intensity ratings were demonstrated between Sessions **0**, **1**, and **3** (ICC [95% CI] = 0.92 [0.86–0.96]). No significant difference in pain intensity was demonstrated between 'high-sensitizers' (mean [95% CI]: 4.5 [3.7–5.3]) and 'low-sensitizers' (mean [95% CI]: 3.9 [3.1–4.6] NRS; unpaired t test: -1.7–0.5, P = 0.263).

**Clinical opiate withdrawal scale.** One subject (#AH39) had a COWS rating at step 2 during the naloxone infusion of 5 (1 at baseline), corresponding to mild opiate withdrawal symptoms. Symptoms included yawning, nausea, restlessness, and anxiousness. No other subjects experienced significant opiate withdrawal symptoms during the infusions.

**Table 4. Mixed-effects model (secondary hyperalgesia area data).**

| Fixed-effect variable | | Estimate (95% CI; cm$^2$) | P-value |
|---|---|---|---|
| Intercept | | 36.1 (1.3–70.9) | 0.042 |
| Intervention | Placebo | -13.4 (-24.2–-2.6) | 0.015 |
| | Naloxone | 0 | - |

Secondary hyperalgesia area (SHA) data from the mixed-effects model with random effect for subject, and fixed effects for the variables 'sensitizers' ('high-sensitizers'/'low-sensitizers'), intervention (naloxone/placebo), target-controlled infusion-step (step 1, 2, 3), Hospital Anxiety and Depression Scale scores, and Pain Catastrophizing Scale scores for the primary outcome SHA.

**Psychometrics.** No subject reached a cut-off score indicating anxiety (HADS-A; median [95% CI]: 3 [2 to 4]) or depressive behavior (HADS-D; 1 [1 to 2]). However, two subjects had a HADS-A score suggestive of an anxiety disorder (#AD03: HADS-A = 8, #AQ09: HADS-A = 9). No subject scored outside the normal range on the PCS score (median [95% CI]: 6 [4–8]).

**Online reaction time test.** A two-way repeated-measures ANOVA showed no significant interaction between the effect of intervention and TCI-step on the reaction time (P = 0.615).

## Adverse events

For a detailed description of adverse events (AEs), see S1 Adverse events.

**Events related to the trial drug.** Adverse events of mild intensity including tiredness, nausea, vomiting, dizziness, headache, itching, and restlessness were experienced by 22/38 subjects receiving naloxone compared to 4/38 subjects following placebo (Chi$^2$ test: P < 0.001). One subject (#AH39) experienced moderate AEs, including anxiety, nausea, and dizziness. One subject (#AX25) experienced anxiety, dizziness, unilateral paresthesia of the arm and leg, and perioral numbness during the naloxone infusion. In both subjects the

**Table 5. Post hoc mixed-effects model (secondary hyperalgesia area data).**

| Fixed-effect variable | | Estimate (95% CI; cm$^2$) | P-value |
|---|---|---|---|
| Intercept | | 46.6 (-3.5–96.8) | 0.068 |
| Intervention | Placebo | -35.4 (-49.8–-21.1) | < 0.001 |
| | Naloxone | 0 | - |
| Sensitizers | 'Low-sensitizers' | -38.1 (-107.6–31.4) | 0.275 |
| | 'High-sensitizers' | 0 | - |
| Time | Session 2 | 49.7 (35.4–64.1) | < 0.001 |
| | Session 4 | 0 | - |
| Intervention*'sensitizer' | Placebo*'low-sensitizer' | 32.2 (12.3–52.2) | 0.002 |
| | Placebo*'high-sensitizer' | 0 | - |
| | Naloxone*'low-sensitizer' | 0 | - |
| | Naloxone*'high-sensitizer' | 0 | - |
| 'Sensitizer'*time | 'Low-sensitizer'*Session 2 | -48.7 (-68.6–-28.7) | < 0.001 |
| | 'Low-sensitizer'*Session 4 | 0 | - |
| | 'High-sensitizer'*Session 2 | 0 | - |
| | 'High-sensitizer'*Session 4 | 0 | - |

Secondary hyperalgesia area (SHA) data from the mixed-effects model with random effect for subject and fixed effects for the variables 'sensitizers' ('high-sensitizers'/'low-sensitizers'), intervention (naloxone/placebo), target-controlled infusion-step (step 1, 2, 3), time (Session 2, 4), Hospital Anxiety and Depression Scale scores and Pain Catastrophizing Scale scores for the primary outcome SHA.

**Table 6. Estimates of fixed effects for the mixed-effects model (secondary hyperalgesia area data).**

| Fixed-effect variable | | Estimate (95% CI; cm$^2$) | P-value |
|---|---|---|---|
| **'High-sensitizers'** | Placebo | -35.4 (-49.8–-21.1) | < 0.001 |
| | Naloxone | 0 | - |
| **'Low-sensitizers'** | Placebo | -3.2 (-17.1–10.7) | 0.651 |
| | Naloxone | 0 | - |

Estimated means of fixed effects from the mixed-effects model using secondary hyperalgesia area (SHA) data. Variables include 'sensitizers' and intervention.

infusion was discontinued prematurely, but both agreed to further trial participation and were not excluded from the analysis.

**Events unrelated to the trial drug.** One subject (#92) experienced a vasovagal syncope immediately following the CHI during Session **0** and was excluded from further participation in the trial (Fig 1). No blistering was observed following the CHIs.

## Protocol violations

Trial drug package #31 was opened unintentionally, and therefore the package was discarded.

One subject (#43) had a body mass index of 30.7 kg/m$^2$ exceeding the predetermined upper margin of 30.0 kg/m$^2$.

The statistical analysis plan concerning the a priori mixed-effects model with random effects for subject was changed before the un-blinding of data. The trial statistician recommended adding the variable time (Session **2**, **4**) to the other fixed effects variables (cf. Materials and methods—Statistical analysis: 'sensitizers'; intervention; TCI-step; HADS-scores, and PCS-scores), since time-dependent changes affecting the analyses were likely to occur.

## Missing values

One subject (#AH39) did not finish the online reaction time test in the last TCI-step due to naloxone-induced side effects. This subject was therefore excluded from the online reaction time analysis.

## Discussion

### Short summary

In the present randomized, controlled, crossover trial, an enriched design was essential in separating sensory phenotypes based on a conditioning heat injury. While the late post-injury naloxone challenge, analyzed by univariate statistics, could not demonstrate reinstatement of latent sensitization, a multivariate mixed-effects model inferred that this could be the case in 'high-sensitizer' subjects.

### Cutaneous heat injury model

**Previous studies.** The CHI model is a reference model in human experimental pain, similarly to the ultraviolet-B (UVB) and capsaicin models [32]. In pharmacodynamical research, the CHI model has demonstrated moderate analgesic efficacy of ketamine [33–36] and opioids [34, 37], but only limited efficacy of anti-inflammatory drugs [38–40], a paradox since the model is inflammatory in nature. The model, however, provides long-lasting hyperalgesia [41].

**Secondary hyperalgesia.** Assessments of SHAs are regularly used as a quantitative outcome measure of central sensitization and has been used in physiological studies [42, 43],

clinical predictive studies [44], and pharmacodynamical studies [19, 35, 45]. Cutaneous heat injury-induced secondary hyperalgesia is a consistent and reproducible measure with low intra-subject variability [20, 46]. Delineation of SHAs is made by punctate (pin-prick) stimulations, conventionally using polyamide monofilaments. However, a comparative trial between the weighted-pin instrument and monofilaments has indicated more reliable measurements, with less variance and larger magnitudes of SHAs with the former method [20].

The development of large SHAs has been revealed as a risk factor for developing chronic pain [47–49]. Further, physiological brain similarities exist between subjects with large SHAs following a CHI and chronic pain patients [18]. On the other hand, an association between SHAs and acute postoperative pain is not readily apparent [50–52]. Although subjects clearly show phenotypical differences in the development of SHAs, the importance of SHAs could not be definitively established in the present proof of concept trial, in terms of opioid receptor-masked latent sensitization.

## Naloxone

**Comparative dose aspects.**   The present trial used a naloxone dose of 3.25 mg/kg, which is 600 to 6,000 times higher than the recommended clinical dose used in the treatment of a severe opioid overdose [53]. Animal studies have unmasked latent sensitization with 0.3–10 mg/kg of naloxone [5, 14] or 3 mg/kg of naltrexone [11, 12]. Uncorroborated data from a positron emission tomography (PET) trial indicated that administration of 0.1 mg/kg of naloxone completely inhibited the binding of $[^{11}C]$-carfentanil to MOR receptors [54]. Given that the naloxone dose in the present trial by far exceeds 0.1 mg/kg, an unknown 'off-target' effect of the drug is not unlikely.

**Adverse events.**   Previous studies have shown that systemic doses up to 6.0 mg/kg have been tolerated well in healthy participants [55–60], and even with higher doses in patients [61–66] with none or only mild to moderate AEs. However, AEs have not been systematically examined in any of these trials. In our previous trial, six out of 15 subjects reported mild AEs, including tiredness, epigastric pain, frontal headache, and photophobia, but no serious adverse events were observed [14]. Similar AEs were observed in the present 38 subjects (cf. Results—Adverse events). Based on these findings, no safety issues regarding the current naloxone dosing have been demonstrated.

**Pharmacokinetics.**   We recently evaluated our three-stage stepwise infusion-algorithm in an exploratory pharmacokinetic study in healthy volunteers (n = 8), where we showed that the naloxone plasma concentrations estimated from the population-kinetic modelling were approximately 3-fold higher than the observed [67]. However, the observed peak plasma concentrations of naloxone and naloxone-3-glucuronide in the eight volunteers [67] were more than 30 times higher than seen in previously published studies [21, 68, 69] and were of the same magnitude (1000 mg/mL) as in the study by Pereira et al. [22].

## Data interpretation

For statistical and methodological reasons, we could not unequivocally demonstrate a statistically significant unmasking effect of naloxone on latent sensitization in the CHI model.

First, univariate and multivariate statistical methods yielded different outcomes. While univariate comparisons did not demonstrate a significant difference between infusion treatments or groups, the post hoc mixed-effects model (including the variable time [cf. Materials and methods—Statistical analysis]) yielded a highly significant effect of naloxone on SHAs in the 'high-sensitizers'. No outlier analysis was stipulated in the protocol, but simple visual inspection of the data reveals high data heterogeneity that may have inferred statistical errors (Fig 6).

However, the assigned P-value 0.01 was targeted at mitigating type 1 errors. In the mixed-effect model, a highly significant interaction between 'sensitizers'*time was observed, indicating that not only sensitizer affiliation, but also the time of injection (Session **2** or **4**) seemed to affect the SHA outcome. We have no definitive explanation for this possible time effect, but likely it is caused by data outliers.

Second, the current and previous human trials investigating the phenomenon of latent sensitization have been restricted to the use of MOR inverse agonists. By contrast, pre-clinical research suggests that several pain inhibitory G-protein coupled receptors, including μ- δ- and κ-opioid receptors, $\alpha_{2A}$-adrenergic receptors, and neuropeptide Y1 and Y2 receptors all contribute to the masking of latent sensitization [5, 11, 12, 70–73]. This raises the question as to whether the mechanisms involved in latent sensitization differ between species, or whether blockade of just opioid receptors is not sufficient to reveal latent sensitization in humans. Indeed, receptor interactions include a 100-fold synergy between endogenous μ-opioid and neuropeptide Y1 receptors [74], leading us to speculate that perhaps a balanced mixture of drugs will be required to reveal latent sensitization in humans.

Third, the conditioning tissue injury used in the present trial, putatively leading to latent sensitization, only covered a fraction of the body surface area ($< 0.1\%$), and only involved a relatively weak inflammatory response, that may have been insufficient to trigger the development of latent sensitization. The CHI model has been back translated to the mouse hind paw (52°C, 40 s). Three weeks later, naloxone dose-dependently reinstated hyperalgesia in these mice. However, the injury was applied to weight-bearing glabrous skin covering a much larger fractional area [75] than in humans. The issue of the limited injury in the present trial could be overcome by studying surgical models with higher severity of tissue injury, e.g., impacted mandibular third molar extraction or groin hernia repair.

Fourth, an issue in previous mouse studies could be a selection bias. The inbred C57BL/6J mouse strain, used in earlier latent sensitization research [11, 12], exhibits markedly enhanced pain sensitivity and thus can be considered belonging to a 'high-sensitizer' clone [76–78]. However, latent sensitization is readily observed in the outbred Sprague-Dawley rat [12].

Addressing some of the above-mentioned issues may accommodate future trials. Nevertheless, the pathophysiological role of latent sensitization in development of chronic pain is speculative. Albeit, if the unmasking of latent sensitization proves to be an essential element of the postsurgical chronification process, it may improve our understanding of the mechanisms involved significantly. This may lead to targeted presurgical, preventive interventions, but also to novel management strategies in established postsurgical pain states.

## Limitations

**Data variability.** The sample size estimates were based on a previous high-dose naloxone trial involving 12 subjects rendering three subjects in each quartile of 'high-sensitizers' and 'low-sensitizers' (cf. Materials and methods—Sample size calculation) [14]. Data from this low-powered trial obviously lead to fragile estimations of sample size. Other important statistical aspects are discussed above.

**Effects during placebo treatment.** The assessments during the placebo infusion yielded substantial SHAs corroborating findings from a previous naloxone trial [13]. Two explanations are possible. First, residual sensitizing effects of the preceding CHI 168 h earlier, enhanced by repeated pin-prick stimuli [41], may occur in susceptible subjects. Second, a 'classic' placebo response may potentially alter the magnitude of secondary hyperalgesia [79]: including subjects' knowledge of the expected dose-dependent effect, and an auditory awareness of changes in the pump infusion rate.

### Strengths

**Trial sample.**   Only healthy young male subjects were included in the trial, providing a very homogenous sample group. Only two subjects were excluded after Session **0**, and missing data were negligible.

**Enrichment design.**   A moderate to substantial agreement between the CHI-sessions was observed regarding the SHAs similar to previous observations [17, 46]. However, there was a significant difference in SHAs between Session **0** and Session **1** for the 'low-sensitizers', which may have been caused by the use of different sites for the CHI across the two sessions (thigh vs. medial calf).

### Methods

Two proficient investigators (ASD, EKJ) carried out all assessments. To reduce carry-over effects between CHI-sessions, contralateral mirror sites were used, including a 'recovery' period $\geq 41$ days between the sessions. The sensory equipment used for the CHIs were regularly calibrated.

### Conclusion

The trial examined naloxone-induced reinstatement of latent sensitization in a cutaneous heat injury model in healthy males. The results could not unequivocally establish the phenomenon of latent sensitization in humans. While latent sensitization may occur in humans, it is not as prevalent a finding as in rodents. Further trials, including postsurgical pain models are needed to prove the clinical significance of latent sensitization and its opposing endogenous analgesia.

### Supporting information

**S1 Checklist. CONSORT 2010 checklist.**
(DOC)

**S1 Table. Trial data.** Table with individual data on demographics, sensitizer allocation, psychometrics (HADS and PCS), secondary hyperalgesia areas in relation to the cutaneous heat injuries and during the target-controlled infusions, VAS pain scores, mechanical pain thresholds, and online reaction times.
(XLSX)

**S1 Protocol. Protocol.**
(PDF)

**S1 Adverse events. Adverse events during the trial.**
(DOCX)

### Acknowledgments

The authors gratefully acknowledge the practical assistance of Emilie Koldborg Jensen, M.B. and Louise Dorothea Skovbjerg, M.B., Neuroscience Center, Copenhagen University Hospitals, Denmark.

### Author Contributions

**Conceptualization:** Bradley Kenneth Taylor, Mads Utke Werner.

**Formal analysis:** Anders Deichmann Springborg, Morten Aagaard Petersen, Theodoros Papathanasiou, Mads Utke Werner.

**Funding acquisition:** Bradley Kenneth Taylor, Mads Utke Werner.

**Investigation:** Anders Deichmann Springborg, Elisabeth Kjær Jensen.

**Methodology:** Mads Kreilgaard, Bradley Kenneth Taylor, Mads Utke Werner.

**Project administration:** Anders Deichmann Springborg, Elisabeth Kjær Jensen.

**Resources:** Anders Deichmann Springborg, Elisabeth Kjær Jensen, Mads Utke Werner.

**Supervision:** Morten Aagaard Petersen, Trine Meldgaard Lund, Bradley Kenneth Taylor, Mads Utke Werner.

**Validation:** Anders Deichmann Springborg, Morten Aagaard Petersen, Theodoros Papathanasiou, Trine Meldgaard Lund, Bradley Kenneth Taylor, Mads Utke Werner.

**Visualization:** Anders Deichmann Springborg, Mads Utke Werner.

**Writing – original draft:** Anders Deichmann Springborg, Mads Utke Werner.

**Writing – review & editing:** Anders Deichmann Springborg, Elisabeth Kjær Jensen, Mads Kreilgaard, Morten Aagaard Petersen, Theodoros Papathanasiou, Trine Meldgaard Lund, Bradley Kenneth Taylor, Mads Utke Werner.

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
