## [Decision Letter · Decision Letter 0]

11 Sep 2020

PONE-D-20-21906

High-dose naloxone: Effects by late administration on pain and hyperalgesia following a human heat injury model.

A randomized, double-blind, placebo-controlled, crossover trial with an enriched enrollment design.

PLOS ONE

Dear Dr. Anders Deichmann Springborg

Thank you for submitting your manuscript to PLOS ONE. After careful consideration, we feel that it has merit but does not fully meet PLOS ONE’s publication criteria as it currently stands. Therefore, we invite you to submit a revised version of the manuscript that addresses the points raised during the review process.

I would appreciate to pay a careful attention in your revision to the reviewer's comments.

We look forward to receiving your revised manuscript.

Kind regards,

Ehab Farag, MD FRCA FASA

Academic Editor

PLOS ONE

Journal Requirements:

2. In your Methods section, please provide additional information about the participant recruitment method and the demographic details of your participants. Please ensure you have provided sufficient details to replicate the analyses such as:   a) a table of relevant demographic details, and b) descriptions of where participants were recruited and where the research took place.

Reviewers' comments:

Reviewer's Responses to Questions

**Comments to the Author**

1. Is the manuscript technically sound, and do the data support the conclusions?

Reviewer #1: Partly

2. Has the statistical analysis been performed appropriately and rigorously? 

Reviewer #1: No

3. Have the authors made all data underlying the findings in their manuscript fully available?

Reviewer #1: Yes

4. Is the manuscript presented in an intelligible fashion and written in standard English?

Reviewer #1: Yes

5. Review Comments to the Author

Reviewer #1: The study entitled ‘High-dose naloxone: Effects by late administration on pain and hyperalgesia following a human heat injury model. A randomized, double-blind, placebo-controlled, crossover trial with an enriched enrollment design’ with the aim to examine the association between injury-induced secondary hyperalgesia and naloxone-induced unmasking of latent sensitization.

This is quite an interesting study but the manuscript requires further improvement.

Comments

Methods

Participants & Enrichment

Page 7 Line 113-114, it was stated ‘Two subjects from Q4 were excluded before Session 1 for reasons unrelated to the trial,’ but in Page 8 Line 129-130, ‘Subjects in Q4 (n = 20) and Q1 (n = 20) continued to experimental Sessions 1 to 4.

Randomization procedure

Page 8 Line 142- 143 & Figure 1, randomization procedure and cross-over design not clear. More information to be provided.

Drugs administration

Page 9-10 Line 165-173 & Page 10-11 175-182, the section to be placed at their respective section write-up.

Page 11 Line 191-196, the section to be placed with Session 2 and 4.

Pain during the cutaneous heat injury

Page 12 Line 223, full name for AUC/s to be provided.

Sample size calculation

Page 13 Line 249, the word ‘high-variance data’ not clear and requires revision.

Statistical analysis

Paired samples t-test and Unpaired samples t test to be written as paired t test and unpaired t test respectively throughout the manuscript.

Page 14 Line 263, the exact type of Wilcoxon test to be stated.

Page 14 Page Line 266-267, ‘time (Session 2, 4 [post hoc cf. Results - Protocol violations])’ requires rewording.

Page 14 Line 268-269. the sentence ‘Factors (P > 0.05), beginning with interactions, were excluded until all included factors attained significance.’ requires fine tuning.

Adjustment on p value/CI if any to be stated. Effect size could be used where applicable.

The order/sequence presentation of the write-up in the methodology requires improvements.

Results

The word Median and 95%CI or Mean and 95% CI to be used throughout the manuscript. For 95%CI, lower ‘to’ upper limit, the word ‘to’ to be replaced with symbol dash’-’

Validity of the enrichment procedure

Page 16 Line 310-311, word median to be stated.

Pairwise comparison

Page 17 Line 324, 326-328, the sentence not clear weather comparing sessions or between groups. The use of Wilcoxon signed rank for what comparison to be clearly stated.

Page 18 Line 334-337 & 345-347, word median (95% CI) and word ‘estimate’ and ‘95% CI’ to be stated.

Page 18 Line 337, statistical test to be denoted.

Mixed-effects model

Page 18 Line 346, results to be placed in table.

Mechanical pain thresholds

Page 19, Line 364, what method use to correcting baseline to be clearly stated.

Page 19 365-366, statistical test to be stated.

Page 20 Line 368-373, mean (95% CI) to be stated where necessary.

No difference to be stated as no significance difference or the difference was not statistically significant.

References to follow journal format.

6. PLOS authors have the option to publish the peer review history of their article (what does this mean?). If published, this will include your full peer review and any attached files.

Reviewer #1: No

---

## [Author Response · Author response to Decision Letter 0]

24 Oct 2020

A. Comments to the Editor concerning ‘Journal Requirements’

1. The authors have ensured that the manuscript meets PLOS ONE style requirements, including those for file naming. 

2. In the Methods section, the authors have provided relevant, additional information about the participant recruitment and the laboratory environment. A table of relevant demographic details has already been provided (Table 3).

B. Comments to the Reviewer

The authors acknowledge with gratitude the effort in preparing the evaluation of the manuscript. Thank you. The authors consider the suggestions and queries for having improved the manuscript significantly.

REVIEWER #1: The study entitled ‘High-dose naloxone: Effects by late administration on pain and hyperalgesia following a human heat injury model. A randomized, double-blind, placebo-controlled, crossover trial with an enriched enrollment design’ with the aim to examine the association between injury-induced secondary hyperalgesia and naloxone-induced unmasking of latent sensitization.

This is quite an interesting study, but the manuscript requires further improvement.

We thank the Reviewer for the compliment, and we will do our best to comply with the comments and queries.

COMMENTS:

Methods: Participants & Enrichment (Q1)

Q1: Page 7 Line 113-114, it was stated ‘Two subjects from Q4 were excluded before Session 1 for reasons unrelated to the trial,’ but in Page 8 Line 129-130, ‘Subjects in Q4 (n = 20) and Q1 (n = 20) continued to experimental Sessions 1 to 4.

R1: We agree that these seeming contradictions may confuse the reader. The sentence: ‘Subjects in Q4 (n = 20) and Q1 (n = 20) continued to experimental Sessions 1 to 4’ refers to the fact that the 20 subjects from the upper quartile (Q4) and the 20 subjects from the lower quartile (Q1) were planned to continue forward to the experimental sessions (1-4). However, after the randomization (prior to participation in Session 1), two subjects were discontinued; thus, no other subjects took their place. The sentence: ‘Two subjects from Q4 were excluded before Session 1 for reasons unrelated to the trial’, is actually from a figure caption, which adds to the confusion. The information in the sections has been updated.

Randomization procedure (Q2)

Q2: Page 8 Line 142- 143 & Figure 1, randomization procedure and crossover design not clear. More information to be provided.

R2: We thank the Reviewer for pointing this out. We have tried to make the information about the randomization procedure clearer.

Regarding Figure 1, this has been updated with information on how many subjects received the allocation. Further, we have made an attempt to make the phases of the enrichment, randomization, and allocation process clearer.

Drugs administration (Q3-Q4)

Q3: Page 9-10, Line 165-173 & Page 10-11 175-182, the section to be placed at their respective section write-up.

R3: The two sections are figure captions, which by PLOS ONE requirements should be placed after the first paragraph where they are cited. We, therefore, believe that the sections are correctly placed.

Q4: Page 11, Line 191-196, the section to be placed with Session 2 and 4.

R4: The section referred to by the Reviewer contains information about the cutaneous heat injury, which is induced in the subjects in Sessions 0, 1, and 3. We, therefore, do not believe that the section should be placed together with Sessions 2 and 4 (drug administration sessions). However, we agree that it is more logical to place the section before information about the drug infusion, since this makes more sense from a chronological aspect. Therefore, the ‘Cutaneous heat injury’ section has been moved to before the ‘Drug administration’ section (Page 9, Line 163-168).

Pain during the cutaneous heat injury (Q5-Q6)

Q5: Page 12, Line 223, full name for AUC/s to be provided.

R5: We thank the Reviewer for pointing this out. The full name for AUC/s has been provided as ‘area under the curve per second’ (Page 12, Line 233).

Sample size calculation (Q6)

Q6: Page 13, Line 249, the word ‘high-variance data’ not clear and requires revision.

R6: We agree that the wording might be misunderstood. Changes have been made to the sentence. 

Statistical analysis (Q8-Q12)

Q7: Paired samples t-test and Unpaired samples t test to be written as paired t test and unpaired t test respectively throughout the manuscript.

R7: We thank the Reviewer for pointing this out. Paired samples t-test and unpaired samples t-test have been corrected to paired t test and unpaired t test, respectively, throughout the manuscript.

Q8: Page 14, Line 263, the exact type of Wilcoxon test to be stated.

R8: The Wilcoxon test has been specified as the Wilcoxon signed-rank test (Page 14, Line 273).

Q9: Page 14 Page Line 266-267, ‘time (Session 2, 4 [post hoc cf. Results - Protocol violations])’ requires rewording.

R9: We agree with the Reviewer that the sentence requires rewording and therefore changes have been made to the sentence.

Q10: Page 14, Line 268-269. the sentence ‘Factors (P > 0.05), beginning with interactions, were excluded until all included factors attained significance.’ requires fine tuning.

R10: We have adjusted the sentence regarding the exclusion of factors in the mixed-effects model. Further, amendments have been made in the Statistical analysis section regarding the mixed-effects model.

Q11: Adjustment on p value/CI if any to be stated. Effect size could be used where applicable.

R11: Regarding the adjustment on p-value/CI, we assume that the Reviewer means if we used any multiple testing correction. We have not used any such corrections. However, a general adjustment on p-value has been made in the sense that a conservative significance level of 0.01 was used. 

Regarding effect size, our co-author biostatistician (MAP) recommends that we, for the mixed-effects model and other statistical analyses, instead use estimates (mean differences) with CI (as we have done) since these are more clinically relevant. 

Q12: The order/sequence presentation of the write-up in the methodology requires improvements.

R12: We agree that the order of sections in the Methods could be improved. As mentioned in R4, we have moved the ‘Cutaneous heat injury’ section to before the ‘Drug administration’ section. 

Results (Q13)

Q13: The word Median and 95%CI or Mean and 95% CI to be used throughout the manuscript. For 95%CI, lower ‘to’ upper limit, the word ‘to’ to be replaced with symbol dash ’-’

R13: We agree that adding mean and median increases the ease of reading. The words median (95% CI) and mean (95% CI) have been added throughout the manuscript. Additionally, ‘to’ has been replaced with the symbol dash ‘–‘ for all confidence intervals. A longer dash was used since some confidence intervals contain negative values and therefore included a short dash ‘-‘ before the number.

Validity of the enrichment procedure (Q14)

Q14: Page 16, Line 310-311, word median to be stated.

R14: The word ‘median [95% CI]’ has been added to the sentence (Page 17, Line 322-323).

Pairwise comparison (Q15-Q17)

Q15: Page 17, Line 324, 326-328, the sentence not clear weather comparing sessions or between groups. The use of Wilcoxon signed rank for what comparison to be clearly stated.

R15: We acknowledge that the sentence may be confusing. Thus, the section has been updated.

Q16: Page 18 Line 334-337 & 345-347, word median (95% CI) and word ‘estimate’ and ‘95% CI’ to be stated.

R16: The Reviewer is correct, and ‘median [95% CI]’ and ‘estimate [95% CI]’ have been added for the first section (Figure 6 caption [Page 18, Line 349-351]). However, for the second section, the numbers (except p-values) have been deleted (Page 18, Line 361-363; cf. R18) 

Q17: Page 18, Line 337, statistical test to be denoted.

R17: The statistical test, Mann-Whitney U test, has been denoted. Additionally, the statistical test has been denoted as Wilcoxon signed-rank test (Figure 6 caption [Page 18, Line 450]).

Mixed-effects model (Q18)

Q18: Page 18 Line 346, results to be placed in table.

R18: By recommendation from our co-author biostatistician (MAP), the results referred to by the Reviewer have been included in a new Table 6 (Page 20, Line 376-379). Further, changes have been made to the 'Mixed-effects model' paragraph (Page 18, Line 360-362)

Mechanical pain thresholds (Q19-Q23)

Q19: Page 19, Line 364, what method use to correcting baseline to be clearly stated.

R19: The method for baseline correction was clarified (Page 20, Line 384-385)

Q20: Page 19 365-366, statistical test to be stated.

R20: We apologize for the mistake and have added the statistical test, Wilcoxon signed-rank test (Page 21, Line 387-388).

Q21: Page 20 Line 368-373, mean (95% CI) to be stated where necessary.

R21: The word ‘mean (95% CI)’ has been stated wherever necessary (Page 21, Line 390-396). 

Q22: No difference to be stated as no significance difference or the difference was not statistically significant.

R22: The appropriate wording has been changed wherever necessary. 

Q23: References to follow journal format.

R23: The corresponding author and the senior author both have perused the references and are not able to retrieve any bibliographical errors or typos. 

For a more detailed response to the Reviewer and Editor's comments, please see the 'Response to Reviewers' letter.

---

## [Editor Report · Decision Letter 1]

28 Oct 2020

High-dose naloxone: Effects by late administration on pain and hyperalgesia following a human heat injury model.

A randomized, double-blind, placebo-controlled, crossover trial with an enriched enrollment design.

PONE-D-20-21906R1

Dear Dr. Anders Deichmann Springborg

We’re pleased to inform you that your manuscript has been judged scientifically suitable for publication and will be formally accepted for publication once it meets all outstanding technical requirements.

Kind regards,

Ehab Farag, MD FRCA FASA

Academic Editor

PLOS ONE
---

## [Editor Report · Acceptance letter]

5 Nov 2020

PONE-D-20-21906R1 

High-dose naloxone: Effects by late administration on pain and hyperalgesia following a human heat injury model.
A randomized, double-blind, placebo-controlled, crossover trial with an enriched enrollment design 

Dear Dr. Springborg:

I'm pleased to inform you that your manuscript has been deemed suitable for publication in PLOS ONE. Congratulations! Your manuscript is now with our production department. 

Kind regards, 

on behalf of

Dr. Ehab Farag 

Academic Editor

PLOS ONE